# Rhabdomyolysis-Induced AKI Was Ameliorated in NLRP3 KO Mice via Alleviation of Mitochondrial Lipid Peroxidation in Renal Tubular Cells

**DOI:** 10.3390/ijms21228564

**Published:** 2020-11-13

**Authors:** Seok Jong Song, Su-mi Kim, Sang-ho Lee, Ju-Young Moon, Hyeon Seok Hwang, Jin Sug Kim, Seon-Hwa Park, Kyung Hwan Jeong, Yang Gyun Kim

**Affiliations:** Division of Nephrology, Department of Internal Medicine, Kyung Hee University College of Medicine, Seoul 05278, Korea; testwar-01@hanmail.net (S.J.S.); miya26@nate.com (S.-m.K.); lshkidney@khu.ac.kr (S.-h.L.); kidmjy@hanmail.net (J.-Y.M.); hwanghsne@gmail.com (H.S.H.); jinsuk0902@hanmail.net (J.S.K.); 01love14@hanmail.net (S.-H.P.)

**Keywords:** rhabdomyolysis, NLRP3, acute kidney injury, myoglobin

## Abstract

Introduction: A recent study showed that early renal tubular injury is ameliorated in Nod-like receptor pyrin domain-containing protein 3 (NLRP3) KO mice with rhabdomyolysis-induced acute kidney injury (RIAKI). However, the precise mechanism has not been determined. Therefore, we investigated the role of NLRP3 in renal tubular cells in RIAKI. Methods: Glycerol-mediated RIAKI was induced in NLRP3 KO and wild-type (WT) mice. The mice were euthanized 24 h after glycerol injection, and both kidneys and plasma were collected. HKC-8 cells were treated with ferrous myoglobin to mimic a rhabdomyolytic environment. Results: Glycerol injection led to increase serum creatinine, aspartate aminotransferase (AST), and renal kidney injury molecule-1 (KIM-1) level; renal tubular necrosis; and apoptosis. Renal injury was attenuated in NLRP3 KO mice, while muscle damage and renal neutrophil recruitment did not differ between NLRP3 KO mice and WT mice. Following glycerin injection, increases in cleaved caspase-3, poly (ADP-ribose) polymerase (PARP), and a decrease in the glutathione peroxidase 4 (GPX-4) level were observed in the kidneys of mice with RIAKI, and these changes were alleviated in the kidneys of NLRP3 KO mice. NLRP3 was upregulated, and cell viability was suppressed in HKC-8 cells treated with ferrous myoglobin. Myoglobin-induced apoptosis and lipid peroxidation were significantly decreased in siNLRP3-treated HKC-8 cells compared to ferrous myoglobin-treated HKC-8 cells. Myoglobin reduced the mitochondrial membrane potential and increased mitochondrial fission and reactive oxygen species (ROS) and lipid peroxidation levels, which were restored to normal levels in NLRP3-depleted HKC-8 cells. Conclusions: NLRP3 depletion ameliorated renal tubular injury in a murine glycerol-induced acute kidney injury (AKI) model. A lack of NLRP3 improved tubular cell viability via attenuation of myoglobin-induced mitochondrial injury and lipid peroxidation, which might be the critical factor in protecting the kidney.

## 1. Introduction

Rhabdomyolysis is caused by the dissolution of skeletal muscle due to various factors, such as trauma, drug toxicity, infection, excessive exercise, and genetic defects [1,2]. Generally, 10–40% of cases of rhabdomyolysis lead to acute kidney injury (AKI) [1,3,4]. Nevertheless, there is no specific treatment targeting the pathogenesis of rhabdomyolysis. Intracellular materials, including myoglobin, electrolytes, and sarcoplasmic proteins, leak into the systemic circulation following rhabdomyolysis [1,3,4]. Among these materials, released myoglobin is freely filtered by glomeruli and reabsorbed by renal tubular cells via endocytosis, and the presence of myoglobin is indicated by red–brown urine [1]. Free myoglobin is not only directly toxic to renal epithelial cells but also acts as a mediator of inflammation. Concentrated myoglobin in renal tubules leads to tubular obstruction when it precipitates with Tamm-Horsfall protein [5]. The binding of oxygen to ferrous oxide derived from myoglobin can lead to the generation of a hydroxyl radical, which can cause mitochondrial damage and tubular apoptosis [6]. Additionally, myoglobin contains a peroxidase-like enzyme that initiates lipid peroxidation and generates isoprostanes [7]. Myoglobin-derived lipid peroxidation and iron accumulation injures the kidney by escalating ferroptosis during rhabdomyolysis [8]. However, there is limited information regarding ferroptosis as a critical pathogenesis of rhabdomyolysis. Damage-associated molecular patterns (DAMPs) can activate NOD-like receptor family proteins and the pyrin domain containing-3 (NLRP3) inflammasome in the absence of pathogens [9,10]. Several studies have suggested that in rhabdomyolysis, myoglobin can mediate AKI by activating the NLRP3 inflammasome [11,12]. One study demonstrated that glycerol-mediated rhabdomyolysis-induced acute kidney injury (RIAKI) is ameliorated in NLRP3 knockout (KO) mice compared to control mice [13]. However, the underlying mechanisms linking NLRP3 to renal injury remain unknown. Renal tubular injury is preceded by immune cell infiltration into the kidney 24 h after glycerol injection, and the depletion of NLRP3 mitigates the renal inflammatory response in a murine RIAKI model [13]. We previously confirmed that renal tubular epithelial cells (TECs) do not express cleaved caspase-1 or release IL-1β [14]. The combination of NLRP3 agonists, such as lipopolysaccharide (LPS), and ATP, nigericin, or monosodium urate fails to induce IL-1β release by renal TECs [15]. Therefore, the depletion of inflammasome-independent NLRP3 in renal TECs could contribute to attenuation of renal tubular injury in glycerol-induced RIAKI. However, there has been no study testing this hypothesis. We reported that NLRP3 depletion decreases mitochondrial damage and apoptosis in hypoxia; in contrast, overexpression of NLRP3 is sufficient to elevate mitochondrial reactive oxygen species (ROS) levels even under normoxic condition [14]. Additionally, when it forms a complex with ASC and caspase-8 in mitochondria, NLRP3 regulates apoptotic cell death in the renal and gut epithelium [15]. This evidence suggests that NLRP3 regulates apoptosis by modulating mitochondrial function in addition to forming the canonical inflammasome. Therefore, we hypothesized that inflammasome-independent NLRP3, especially in renal tubular cells, is involved in mediating renal injury in RIAKI and we attempted to clarify the precise mechanism underlying myoglobin-induced apoptosis and ferroptosis.

## 2. Results

### 2.1. NLRP3 KO Mice Were Protected from RIAKI

Glycerol-mediated RIAKI was established in wild-type (WT) and NLRP3 KO mice. Serum creatinine levels were increased 24 h after glycerol injection in mice of both genotypes and were lower in NLRP3 KO mice than in WT mice (Figure 1A). However, RIAKI-induced elevation of aspartate aminotransferase (AST) and alanine transferase (ALT) was not significantly different between WT and NLRP3 KO mice, suggesting that NLRP3 depletion did not prohibit glycerol-induced muscle damage but protected the kidney from RIAKI (Figure 1B,C). Periodic acid-schiff (PAS) and terminal deoxynucleotidyl transferase dUTP nick end labeling (TUNEL) staining showed that glycerol-induced tubular necrosis and apoptosis were decreased in NLRP3 KO mice compared to WT mice (Figure 1D,F,G,I). However, there was no difference in the number of Ly6B-positive intrarenal cells between WT and NLRP3 KO mice (Figure 1E,H). NLRP3 expression was increased in the kidneys of mice with RIAKI compared to control mice (Figure 2A,B). Kidney injury molecule-1 (KIM-1) and neutrophil gelatinase associated lipocalin (NGAL), markers of proximal tubular injury, were augmented in the kidneys of WT mice with RIAKI, whereas this increase was significantly mitigated in the kidneys of NLRP3 KO mice with RIAKI. In particular, in the kidneys of mice with RIAKI, glutathione peroxidase 4 (GPX4) was downregulated, showing that ferroptosis was induced. Interestingly, the expression of ferroptotic markers was decreased in the kidneys of NLRP3-depleted mice with RIAKI compared with those of WT mice with RIAKI. The expression of apoptotic markers, such as cleaved caspase-3 and poly (ADP-ribose) polymerase (PARP), was elevated in the kidney following glycerol injection, whereas NLRP3 depletion reduced the expression of these markers. NLRP3 depletion decreased the mRNA expression of inflammatory cytokines, including TNF-α, IL-1β, and IL-6, induced by rhabdomyolysis (Figure 2C–E).

### 2.2. NLRP3 Depletion in Renal Tubular Cells Mitigated Myoglobin-Induced Ferroptosis and Apoptosis

In vivo studies suggest that the renal protective effect of NLRP3 depletion mainly originates from alleviation of renal tubular injury; thus, experiments were performed on tubular cells. Ferrous myoglobin was applied to HKC-8 cells to induce an in vitro model of rhabdomyolysis. Since reduced myoglobin (Fe^2+^) has been reported to be cytotoxic, ascorbic acid was added to metmyoglobin (Fe^3+^) to produce ferrous myoglobin [16]. Cell viability decreased as the myoglobin concentration increased to 5 mg/mL, and approximately 60% of 5 mg/mL myoglobin-treated cells survived (Figure 3A). At 1, 5, and 10 mg/mL, ferrous myoglobin similarly increased NLRP3 expression in HKC-8 cells. Since ferroptosis and apoptosis in the kidney are the primary manifestations of RIAKI, the expression of markers related to these forms of cells death were analyzed in HKC-8 cells. Ferrous myoglobin induced an increase in ACSL4 expression, a decrease in GPX4 expression and upregulation of cleaved caspase-3 and PARP expression in HKC-8 cells (Figure 3B,C). This result suggested that rhabdomyolysis-mediated renal tubular cells underwent ferroptosis and apoptosis. These changes were reversed in siNLRP3-treated HKC-8 cells, although the level of GPX4 was not significantly altered (Figure 3D,E).

### 2.3. NLRP3 Depletion in Renal Tubular Cells Attenuated Myoglobin-Induced Mitochondrial Injury

Previously, we showed that inflammasome-independent NLRP3 regulates apoptosis in TECs by interacting with mitochondria and mediating mitochondrial ROS production [14]. Therefore, we decided to examine whether mitochondrial damage under myoglobin stimulation varies depending on the presence or absence of NLRP3. The mitochondrial membrane potential (JC-1) was markedly reduced following ferrous myoglobin stimulation, and this decrease was reversed in siNLRP3-treated HKC-8 cells (Figure 4A,B). Myoglobulin stimulation increased the expression of the mitochondrial fission protein dynamin-related protein 1 (DPR1) and decreased the expression of the mitochondrial fusion protein Mitofusin 1 (MFN1) (Figure 4C,D). Additionally, the mitochondrial changes were attenuated in siNLRP3-treated HKC-8 cells.

### 2.4. NLRP3 Depletion Contributed to Decreasing Myoglobin-Induced ROS Levels and Lipid Peroxidation

Next, we evaluated ROS and lipid peroxidation levels in myoglobin-induced renal tubular cells. BODIPY staining was increased, and BODIPY was colocalized with MitoTracker in HKC-8 cells following myoglobin administration (Figure 5A). Myoglobin-induced lipid peroxidation was significantly attenuated in siNLRP3-treated HKC-8 cells compared to ferrous myoglobin-treated HKC-8 cells. Malondialdehyde (MDA) production was also augmented in ferrous myoglobin-stimulated HKC-8 cells, whereas it was significantly decreased in NLRP3-depleted HKC-8 cells (Figure 5B). The total ROS concentration, as determined by DCFH-DA, was increased in myoglobin-stimulated HKC-8 cells and markedly reduced in siNLRP3-treated HKC-8 cells (Figure 5D). 4-hydroxynonenal (4-HNE) expression was increased in response to myoglobin and decreased in siNLRP3-treated HKC-8 cells (Figure 5E,F). In particular, 4-HNE was upregulated following NLRP3 pCMV6-induced NLRP3 overexpression, even in the absence of stimulation. These results suggested that NLRP3 could be involved in regulating lipid peroxidation in renal tubular cells under myoglobin stimulation.

## 3. Discussion

In this experiment, we demonstrated the following: (1) renal damage caused by rhabdomyolysis is mitigated in NLRP3-depleted mice compared to WT mice via alleviation of renal inflammation, apoptosis, and ferroptosis despite no difference in muscle damage; (2) myoglobin-induced apoptosis and lipid peroxidation are attenuated in NLRP3-depleted tubular cells; (3) depletion of NLRP3 in renal tubular cells improves mitochondrial damage and alters mitochondrial biogenesis induced by myoglobin; and (4) the preservation of mitochondria following myoglobin simulation might contribute to alleviating lipid peroxidation. Several studies have suggested that NLRP3 can aggravate renal injury in the pathogenesis of RIAKI; nevertheless, the precise mechanisms are not yet clear [11,13]. Our study suggested that reduction in mitochondrial damage and lipid peroxidation in renal tubules is the main contributor to the protection of the kidneys in NLRP3 KO mice with RIAKI. NLRP3 is known to mediate renal injury via inflammasome formation in various renal diseases, such as RIAKI, ischemic reperfusion injury (IRI), contrast-induced AKI, and unilateral ureter obstruction nephropathy [17,18,19]. These renal diseases can be ameliorated by NLRP3 inflammasome inhibitors or genetic deletion of inflammasome components [20,21,22]. However, it has been shown that caspase-1 inhibitors do not improve renal tubular apoptosis in rats with RIAKI [11]. Additionally, tubular apoptosis following IRI is decreased in NLRP3 KO mice compared with WT mice, whereas WT mice engrafted with NLRP3 KO bone marrow fail to show improvement [17]. Previously, we confirmed that in renal tubular cells, NLRP3 regulates mitochondrial damage [14]. Similarly, NLRP3 KO renal tubular cells attenuate apoptosis in mitochondria caused by TNFα/CHX [15]. Consistently, our data elucidated that the absence of NLRP3 in renal tubular cells mitigates tubular apoptosis and lipid peroxidation in response to myoglobin stimulation. The fundamental cause of this protective effect was assumed to be connected to the preservation of mitochondria. Tubular cell damage without an increase in the number of intrarenal immune cells is the main manifestation during the early phase of RIAKI [13]. Thus, the elimination of NLRP3 from renal tubular cells could be a critical factor in improving renal function via mitochondrial protection during early RIAKI. The results showing that myoglobin-induced apoptosis and lipid peroxidation were mitigated in NLRP3-depleted renal tubular cells supported our suggestion. Our experiments also demonstrated that glycerol injection led to an increase in the levels of inflammatory cytokines, such as TNFα, IL-β, and IL-6, in the mouse kidney. Inflammation is the main pathogenic feature of AKI induced by rhabdomyolysis, and previous studies have shown that pattern recognition receptors such as TLR4 and NLRP3 are upregulated in the kidney in RIAKI [13,23,24]. We found that inflammatory cytokine levels were decreased in the kidneys of NLRP3-depleted mice compared with those of WT mice, which is in line with a previous study [13]. Nevertheless, the number of intrarenal immune cells were not increased in the kidney 24 h following glycerol injection. Therefore, we assume that the main factor in decreasing inflammation is attenuation of inflammatory responses in renal tubular cells rather than renal immune cells. NLRP3 in nonimmune cells is involved in potentiating mitochondrial ROS levels and ultimately augmenting fibrosis under TGF-β stimulation [25,26]. In addition, NLRP3 KO fibroblasts recruit more anti-inflammatory macrophages than inflammatory macrophages under TGF-β stimulation, ultimately leading to a decrease in inflammation [27]. These data suggest that the absence of NLRP3 in nonimmune renal resident cells may be a principal factor in attenuating renal inflammation during the early phase of RIAKI. This study showed that ferroptosis occurs in glycerol-induced AKI, as ASCL4 expression was upregulated and GPX4 expression was downregulated in the glycerol-injected kidney. Two main factors that lead to ferroptosis are an inability to reduce lipid peroxide levels due to the absence or inactivation of GPX4 and increased formation of lipid peroxides [28]. Oxyferrous myoglobin is converted to metmyoglobin (MetMb, Fe^3+^) via auto-oxidation and can catalytically accelerate lipid peroxidation [29]. The administration of antioxidant agents preserves renal function in glycerol-induced rhabdomyolysis mice [30,31]. Ferroptosis inhibitors effectively ameliorate renal structure and function, while the inhibition of apoptosis or necroptosis does not inhibit renal injury in glycerol-induced AKI [8]. Our study demonstrated that NLRP3 depletion decreased lipid peroxidation and ferroptosis in the kidney during the early phase of RIAKI. The levels of lipid peroxides, including MDA and 4-HNE, and ferroptosis inducers, such as ASCL4, were reduced in NLRP3-deficient renal tubular cells in response to myoglobin stimulation, whereas GPX4 expression was not changed. It is not clear how NLRP3 absence contributes to inhibiting lipid peroxidation. In our study, NLRP3-depleted tubular cells mitigated mitochondrial dysfunction and enhanced mitochondrial biogenesis. Phospholipids in mitochondrial membranes are the primary targets of ROS attack, which results in lipid peroxidation [32]. ROS produced by rhabdomyolysis may extend to form oxidized mitochondrial lipids and are connected to mitochondrial dysfunction. Myoglobin-induced mitochondrial dysfunction and altered mitochondrial biogenesis were reversed when NLRP3 was depleted. Therefore, we speculate that decreases in mitochondrial dysfunction and lipid peroxide levels following NLRP3 depletion can lead to a decrease in the induction of ferroptosis. Correspondingly, NLRP3 overexpression caused an increase in 4-HNE levels, even in the absence of stimulation. However, iron-dependent ferroptosis induced by myoglobin might not be fully blocked in NLRP3-deficient tubular cells, and, thus, GPX4 expression was not decreased. Recent evidence suggests that ACSL4 and activated NLRP3 are similarly localized in the mitochondria-associated membrane, which is located at the junction between the endoplasmic reticulum and mitochondria [33,34]. Therefore, more research is needed to determine whether the interaction between NLRP3 and ACSL4 is connected to the regulation of lipid peroxidation and ferroptosis. This study has limitations. We used conventional NLRP3 KO mice in the animal experiments since tubular cell-specific NLRP3 KO mice were not available. Thus, we cannot thoroughly conclude that this renal protective effect originates solely from the depletion of inflammasome-independent NLRP3 in renal tubular cells. Further studies using cell-specific KO mice and NLRP3 inhibitors are necessary for clarification. In addition, we could not clarify the mechanisms by which myoglobin stimulates renal tubular cells in this study. Nevertheless, this study confirmed for the first time that ferrous myoglobin upregulated NLRP3 expression in renal tubular cells. In summary, NLRP3 depletion during rhabdomyolysis protects against renal injuries via attenuation of myoglobin-induced mitochondrial dysfunction and subsequent lipid peroxidation in renal tubular cells. This study shows that ferroptosis may be the critical factor in inducing renal injury in RIAKI and that inflammasome-independent NLRP3 in renal tubules is associated with the regulation of lipid peroxidation. NLRP3 may be a new therapeutic target for rhabdomyolysis.

## 4. Material and Methods

### 4.1. Animal Models

All animal experiments were performed according to the guidelines, care, and use of Experiments Animals Committee of Kyunghee University Hospital at Gangdong (approval number: KHNMC AP2018-004, 1 February 2018). C57BL/6J WT mice were purchased from Jun Bio.inc. (Suwon, Korea). RGEN/Cas9 NLRP3 KO mice were established from Macrogen (Seoul, Korea). The establishment of NLRP3 KO mice was precisely described in the previous study [14]. The animals were maintained on a 12 h dark/light cycle with free access to food and water. For the purpose of inducing rhabdomyolysis, the two hind limbs of the animals were intramuscularly injected with 5 mL/kg of 50% glycerol in 8–10-week-old male mice. Water was withheld for 24 h before the glycerol injection to increase the incidence of renal injury.

### 4.2. Blood Chemistry

Serum levels of creatinine, blood urea nitrogen (BUN), AST, and ALT were measured by using Vet test 8008 (IDEXX, Ludwigsburg, Germany) according to the manufacturer’s instructions.

### 4.3. Histopathology

Kidneys were fixed in 10% neutral buffered formalin, embedded in paraffin, cut into 4 μm sections, and stained with PAS reagent. Tissue sections were viewed by a light microscope at ×200 magnification. For semiquantitative analysis, 10 different fields at the corticomedullary junction from each group were randomly selected. Necrotic tubules were scored on a scale of 0 to 5 based on the percentage of necrosis area as follows: 0, absent; 1, 1–25%; 2, 26–50%; 3, 51–75%; 4, 76–99%; and 5, 100%.

### 4.4. TdT Mediated dUTP Nick End Labeling (TUNEL) Assay

Apoptosis in renal tissues was identified by TUNEL assay with an in situ Cell Death Detection kit following the manufacturer’s instructions (Roche Applied Science, Indianapolis, IN, USA). The number of apoptotic cells were counted under a fluorescence microscope at ×200 magnification. At least ten areas at the corticomedullary junction in the sections from different mice of each group were determined and averaged.

### 4.5. Real-Time PCR

Total RNA was extracted from renal tissues using a QIAzol^®^Lysis Reagent (QIAGEN, Hilden, Germany) according to the manufacturer’s instructions. The RNA concentration and purity were confirmed with Nanodrop 2000. Real-time RT-PCR analysis was performed using the Step One Plus Real-Time PCR System (Applied Biosystems, Foster City, CA, USA) to detect mRNA expression of IL-6, TNF-α, and IL-1β. Each sample was processed in duplicate in separate tubes to quantify target gene expression and the results were normalized to 18S expression.

### 4.6. Western Blot

Kidney tissue and cells were lysed in an ice-cold lysis buffer (PRO-PREP™ Protein Extraction solution, INTRON, Seongnam-si, Korea). Proteins were separated with 10% PAGE and electroblotted onto a PVDF membrane (BIO-RAD, Seoul, Korea). The membrane was incubated with primary antibody raised against NLRP3 (NOVUS, Centennial, CO, USA), KIM-1 (Abcam, Cambridge, MA, USA), NGAL (Abcam, Cambridge, MA, USA), ACSL4 (Santa cruz, Dallas, TX, USA), GPX4 (Abcam, Cambridge, MA, USA), Cleaved caspase-3 (Cell signaling, Danvers, MA, USA), PARP (Cell sinaling, Danvers, MA, USA), dynamin-1-like protein (DRP1) (Abcam, Cambridge, MA, USA), Mitofusin1 (Abcam, Cambridge, MA, USA), β-actin (Santa cruz, Dallas, TX, USA), GAPDH (Cell Signaling, Danvers, MA, USA) (1:1000) and, subsequently, with horseradish peroxidase-conjugated goat anti-rabbit or mouse immunoglobulin G (1:10,000, BETHYL). The immunoreactive bands were detected by chemiluminescence (ECL, Advansta, CA, USA). β-actin and GAPDH were used as internal controls of cells and tissues.

### 4.7. Cell Culture

The human renal proximal tubular epithelial cell line HKC8 was obtained from Dr. L. Rausen (Johns Hopkins University, Baltimore, MD, USA) and was maintained in Dulbecco’s Modified Eagle Medium supplemented with Ham’s F12 medium (DMEM/F12; Thermo Fisher Scientific, Waltham, MA, USA). DMEM/F12 was supplemented with 10% fetal bovine serum and 1% penicillin/streptomycin (WelGENE, Daegu, Korea). Myoglobin from equine heart muscle (M5696, Sigma, St. Louis, MO, USA) and 2 mM ascorbic acid (Sigma, St. Loiuse, MO, USA) was dissolved in medium and prepared with ferrous myoglobin right off before in vitro experiment. The concentration of myoglobin was adjusted to 200 mM with ascorbic acid 2 mM.

### 4.8. Small Interfering RNA Knockdown Experiments

Duplex small interfering RNAs (siRNAs) targeting NLRP3 (ORIGENE, Rockville, MD, USA) and a control siRNA were purchased from Bioneer Inc. (Seoul, Korea). HKC8 cells were transfected using Lipofectamine 2000 (Invitrogen, Waltham, MA, USA), after which these cells were utilized for functional studies 24 h later, and knockdown efficiency was assessed by Western blot analysis using NLRP3 and GAPDH antibodies.

### 4.9. MTT Assay

After myoglobin (with or without L-ascorbic acid 2 mM) treatment, the cell viability of HKC8 cells plated in 24-well plates was detected by using 3-(4,5-dimethylthiazol-2yl)-2,5-diphenyltetrazolium bromide (MTT) assay. Briefly, the cells were treated with MTT and incubated at 37 °C for 4 h. The supernatant was removed, and the dye was dissolved with 200 µL of DMSO and shaken on an orbital shaker for 10 min in the dark. The optical density (OD) was recorded at 570 nm using a spectrophotometer. Each experiment was performed in triplicate.

### 4.10. Flow Cytometric Analysis

Mitochondrial membrane potential Δ*Ψ*_m_ was measured by the sensitive and relatively mitochondrion-specific lipophilic cationic probe fluorochrome 5,5′,6,6′-tetrachloro-1,1′,3,3′-tetraethylbenzimidazoly-carbocyanine iodide (JC-1) (Molecular Probes, Eugene, OR, USA). Briefly, HKC8 cells were incubated with JC-1 (5 µmol/L) at 37 °C for 20 min and examined by flow cytometry (BD FACSCaliber Flow Cytometry, San Jose, CA, USA). Intracellular ROS was measured by 2′,7′-dichlorofluorescin diacetate (DCF-DA) (Sigma, St. Louis, MO, USA). Briefly, HKC8 cells were incubated with DCF-DA (5 µM) at 37 °C for 30 min and examined by flow cytometry (BD FACSCaliber Flow Cytometry, San Jose, CA, USA).

### 4.11. Immunocytochemistry

Cells were fixed with 4% paraformaldehyde after being stained to BODIPY (Thermo Fisher Scientific, Waltham, MA, USA) and Mitotracker (Thermo Fisher Scientific, Waltham, MA, USA). The nuclei were stained with DAPI (1:1000, Invitrogen). The slides were mounted with Fluorescence Mounting Medium (Dako), and images were acquired by a confocal microscope (Carl Zeiss LSM 700, Jena, Germany). Z-stack of images were projected into one plane (maximum intensity projection).

### 4.12. Lipid Peroxidation Product Assay

Lipid peroxidation product malondialdehyde (MDA) in HKC-8 cells (2 × 10^6^ cells) was measured using a commercial MDA kit (Abcam, Cambridge, MA, USA). The spectrophotometric absorbance was assessed at 532 nm in accordance with the manufacturer’s instructions.

### 4.13. Statistical Analysis

Statistical analyses were conducted using SPSS software (version 20 SPSS, Inc., Chicago, IL, USA) and GraphPad Prism 5.0. Descriptive data were presented as mean ± SEM. Results were analyzed using the Kruskal–Wallis nonparametric test for multiple comparisons and Mann–Whitney test for two objects; *p* values < 0.05 were considered statistically significant.

## Figures and Tables

**Figure 1 ijms-21-08564-f001:**
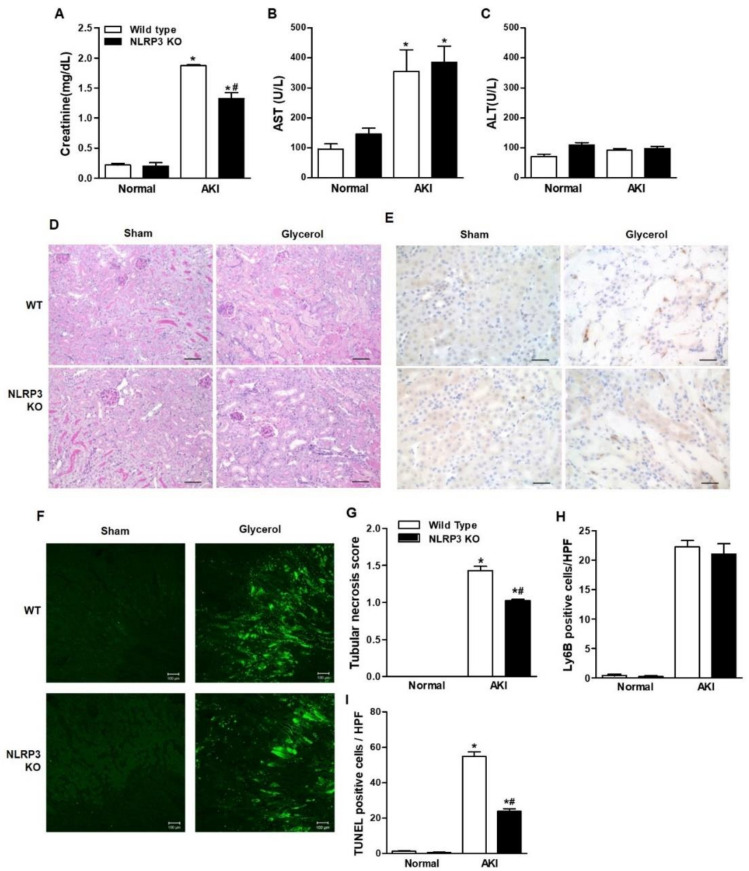
Glycerol-induced renal injury was attenuated in NLRP3 KO mice. Glycerol was intramuscularly injected into WT and NLRP3 KO mice, and the mice were euthanized 24 h after injection. (**A**–**C**) The plasma levels of creatinine, AST, and ALT were measured. (**D**) PAS-stained (200× magnification), (**E**) Ly6B-stained (400× magnification), (**F**) TUNEL-stained renal cortices of sham and glycerol-injected WT and NLRP3 KO mice (200× magnification). (**G**) Tubular necrosis was scored in 10 randomly selected fields. (**H**) Ly6B-positive cells were counted in high-power fields. (**I**) Green fluorescent TUNEL-positive puncta were counted. We used 6 mice per group. NLRP3, NOD-like receptor family proteins and the pyrin domain containing-3; AST, aspartate aminotransferase; ALT, alanine transferase; PAS, periodic acid-schiff; TUNEL, TdT-mediated dUTP nick end labeling, * *p* < 0.05 vs. sham mice, # *p* < 0.05 vs. WT mice.

**Figure 2 ijms-21-08564-f002:**
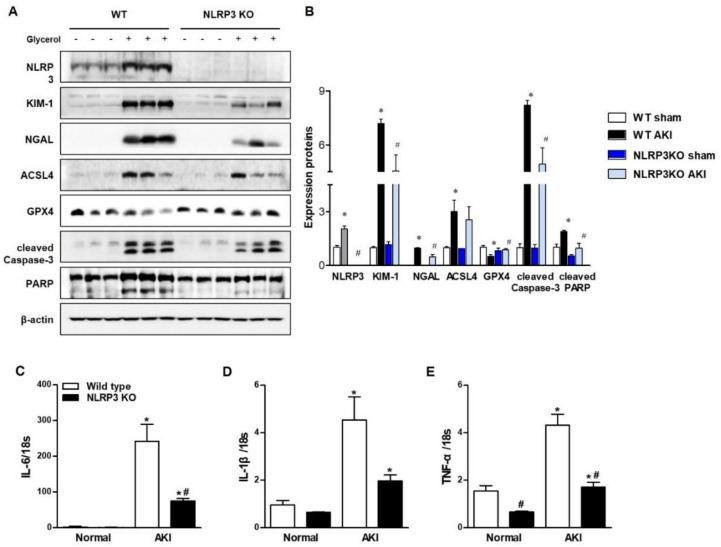
RIAKI led to increases in tubular injury, apoptosis, ferroptosis, and inflammation in the kidney, which were abrogated in the kidneys of NLRP3 KO mice. (**A**) Western blot analysis of NLRP3, KIM-1, NGAL, ACSL4, GPX4, cleaved caspase-3, and PARP expression in the kidneys of sham and glycerol-injected WT and NLRP3 KO mice. (**B**) Densitometric analysis of the Western blots. (**C**–**E**) RT-PCR analysis of IL-6, IL-1β, and TNF-α expression in sham and glycerol-injected WT and NLRP3 KO mice. KIM-1, kidney injury molecule-1; NGAL, neutrophil gelatinase associated lipocalin; ACSL4, acyl-CoA synthetase long-chain family; GPX4, glutathione peroxidase 4; PARP, poly (ADP-ribose) polymerase; IL-6, interleukine-6; IL-1β, interleukine-1β; TNF-α, tumor necrosis factor-α. We used 6 mice per group. * *p* < 0.05 vs. sham mice, # *p* < 0.05 vs. WT mice.

**Figure 3 ijms-21-08564-f003:**
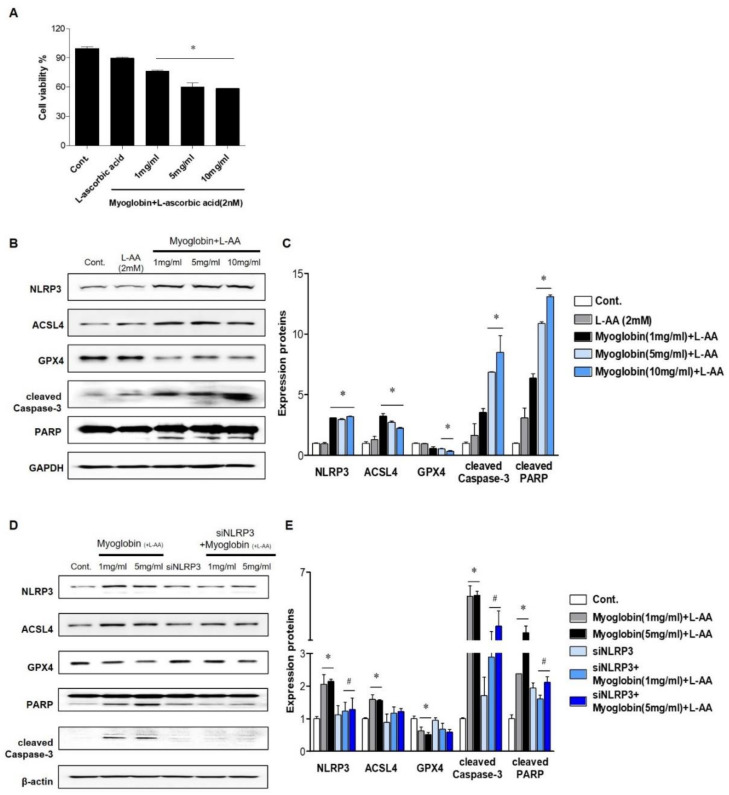
NLRP3-depleted HKC-8 cells were protected from ferrous myoglobin-induced apoptosis and ferroptosis. (**A**) HKC-8 cells were treated with various doses of myoglobin for 24 h, and cell viability was measured by the MTT assay. (**B**,**C**) The expression of NLRP3, ASCL4, GPX4, cleaved caspase-3, and PARP in HKC-8 cells treated with different doses of myoglobin was analyzed by immunoblotting and densitometric analysis. (**D**,**E**) Western blot analysis of NLRP3, ASCL4, GPX4, cleaved caspase-3, and PARP expression in HKC-8 cells transfected with or without siNLRP3 and treated with or without 1 mg/mL or 5 mg/mL myoglobin and densitometric analysis. We repeated 3 experiments in the respective sections. * *p* < 0.05 vs. Mb-untreated cells, # *p* < 0.05 vs. siNLRP3-untreated cells.

**Figure 4 ijms-21-08564-f004:**
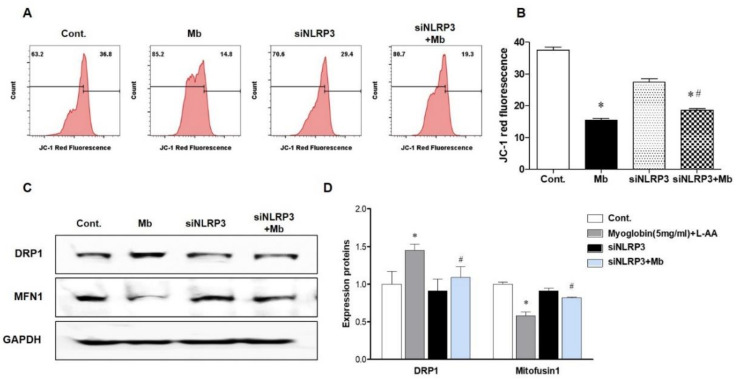
NLRP3-depleted HKC-8 cells were protected from myoglobin-induced mitochondrial injury. HKC-8 cells transfected with or without siNLRP3 were treated with control or myoglobin for 24 h. (**A**,**B**) Flow cytometry analysis of JC-1-stained cells for the detection of mitochondrial membrane potential changes and graphs showing fluorescence. (**C**,**D**) Western blot pictures showing DRP1 and MFN1 expression and densitometric analysis. DRP1, dynamin-related protein 1; MFN1, mitofusin 1. We repeated 3 experiments in the respective sections. * *p* < 0.05 vs. Mb-untreated cells, # *p* < 0.05 vs. siNLRP3-untreated cells.

**Figure 5 ijms-21-08564-f005:**
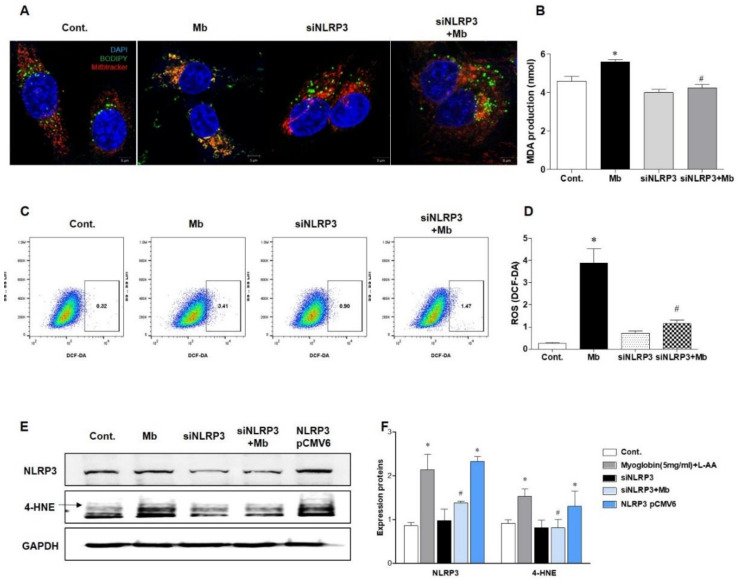
NLRP3-depleted HKC-8 cells were protected from myoglobin-induced ROS production. HKC-8 cells transfected with or without siNLRP3 were treated with control or myoglobin for 24 h. (**A**) Confocal microscopy (63× magnification) and (**B**) analysis of MDA production for the evaluation of lipid peroxidation induced by myoglobin in HKC-8 cells. (**C**,**D**) Flow cytometry of DCFH-DA-stained cells for the detection of ROS production induced by myoglobin in HKC-8 cells. (**E**,**F**) Immunoblotting for NLRP3 and 4-HNE in HKC-8 cells and densitometric analysis. MDA, malondialdehyde; 4-HNE, 4-hydroxynonenal. We repeated 3 experiments in the respective sections. * *p* < 0.05 vs. Mb-untreated cells, # *p* < 0.05 vs. siNLRP3-untreated cells.

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
