# Peer review of "Rhabdomyolysis-Induced AKI Was Ameliorated in NLRP3 KO Mice via Alleviation of Mitochondrial Lipid Peroxidation in Renal Tubular Cells"

_ijms, 2020, doi:10.3390/ijms21228564_

Round 1

Reviewer 1 Report

The authors reported the effect of NLRP3 on rhabdomyolysis-induced AKI with studies using mice and renal tubular cells. The date was examined enough, and manuscript was written well.

The authors examined and discussed about myoglobin induced NLRP3 inflammasome. They should discuss how the myoglobin is uptaken in renal tubular cells before reaction with NLRP3.

The authors should discuss about the role of inflammasome-independent NLRP3 reaction in this disease setting.  

Author Response

Reviewer 1

We appreciate the reviewer’s time and effort spent evaluatint our original submission. We also thank the reviewer for the helpful and insightful comments, which we belive have helped us improve our manuscript. The specific comments are cited in bold type and are followed by our detailed responses. The text in the revised manuscript is shown in red.

The authors reported the effect of NLRP3 on rhabdomyolysis-induced AKI with studies using mice and renal tubular cells. The date was examined enough, and manuscript was written well.

The authors examined and discussed about myoglobin induced NLRP3 inflammasome. They should discuss how the myoglobin is uptaken in renal tubular cells before reaction with NLRP3.

Response: Thank you for your valuable comment. Our experiment confirmed for the first time that ferrous myoglobin upregulated NLRP3 expression in renal tubular cells. Unfortunately, the mechanisms by which NLRP3 agonists stimulate renal tubular cells have not yet been clarified. Further study is needed for clarification. We added this comments in the discussion part.

In addition, we could not clarify the mechanisms by which myoglobin stimulates renal tubular cells in this study. Nevertheless, this study confirmed for the first time that ferrous myoglobin upregulated NLRP3 expression in renal tubular cells.

The authors should discuss about the role of inflammasome-independent NLRP3 reaction in this disease setting.  

Response: We appreciate your critical feedback. Inflammatory cell recruitment was not observed during the early phase of glycerol-induced AKI in our study. In line with our findings, previous study identified rare immune cell infiltration in the renal cortex and suggested that injuries in renal tubular cells might be the main pathogenesis rather than immune cell infiltration during early RIAKI (Sci Rep 2015:5:10901). It was assumed that the improvement could originate from the absence of NLRP3 in renal tubular cells. The results showing that myoglobin-induced apoptosis and lipid peroxidation were mitigated in NLRP3-depleted renal tubular cells supported our suggestion. However, many studies, including our study, confirmed that renal tubular cells could not form inflammasomes despite they having a substantial amount of NLRP3. Therefore, inflammasome-independent NLRP3 in renal tubules is thought to the critical factor in the progression of renal injuries in RIAKI. We added one sentence in the discussion part.

The results showing that myoglobin-induced apoptosis and lipid peroxidation were mitigated in NLRP3-depleted renal tubular cells supported our suggestion.

In addition to the points you made, we added one reference numbered 24 (KRCP 2018;37(3):183-196) in the discussion part for clarification.

Reviewer 2 Report

Scientific question

Due to the frequency of AKI investigations onto the pathogenesis of the condition are needed. Investigating rhabdomyolysis induced AKI reduced the types of AKI to which this work is relevant but there is still clinically relevant scientific question here.

Abstract

A clear synopsis of background to the work presented, along with a succinct outline of the work and reasonable conclusions.

Methods

Indications of the number of animals included in the study would have been informative because the number of mice in each of the experiments is not stated in the legends for figures 1 and 2.

Results

The experiments used to address the scientific question are well thought out.

Use of NLRP3 KO mice in this study has its limitations but they are acknowledged by the authors in the discussion.

An indication of the number of experimental repeats for each of the HKC-8 cell experiments would have been useful.

Figure 1 D and E are both stated to be x200 magnification but I do not think this is correct. The cell and nuclei size differ.

Figure 2B spelling NLRP3KO SKI should be NLRP3KO AKI

Figure 2B – ACSL4 increased in WT AKI. This is also seen in the NLRP3KO AKI but in the text says that there is decreased expression of marker of ferroptotic markers in the NLRP3KO AKI mice compared to the WR AKI.

Figure 3 C and E, it is difficult to which sets of data on the graphs are significantly different, maybe the size of the graphs should be increased and the size of the western blots should be decreased proportionately.

Figure 5 B and the legend -  in the figure # is vs control but in the legend # is vs myoglobin+L-AA, and in the Figure *is vs myoglobin+L-AA and in the legend * is vs control.

The discussion is wide and detailed and clearly lays out where this work fits in the field.

Author Response

Reviewer 2

We appreciate the reviewer’s time and effort spent evaluatint our original submission. We also thank the reviewer for the helpful and insightful comments, which we belive have helped us improve our manuscript. The specific comments are cited in bold type and are followed by our detailed responses. The text in the revised manuscript is shown in red.

Scientific question

Due to the frequency of AKI investigations onto the pathogenesis of the condition are needed. Investigating rhabdomyolysis induced AKI reduced the types of AKI to which this work is relevant but there is still clinically relevant scientific question here.

Abstract

A clear synopsis of background to the work presented, along with a succinct outline of the work and reasonable conclusions.

Methods

Indications of the number of animals included in the study would have been informative because the number of mice in each of the experiments is not stated in the legends for figures 1 and 2.

Response: We appreciate your valuable comment. A total 24 mice were used, and the respective numbers were 6 for sham WT, 6 for sham KO, 6 for AKI WT, and 6 for AKI KO. We added this information to the methods section.

A total 24 mice were used, and the respective numbers were 6 for sham WT, 6 for sham NLRP3 KO, 6 for AKI WT, and 6 for AKI NLRP3 KO.

Results

The experiments used to address the scientific question are well thought out.

Use of NLRP3 KO mice in this study has its limitations but they are acknowledged by the authors in the discussion.

An indication of the number of experimental repeats for each of the HKC-8 cell experiments would have been useful.

Reponse: Thank you for your feedback. We repeated 3 experiments in the respective sections. We added this information to the methods section.

We repeated 3 experiments in the respective sections.

Figure 1 D and E are both stated to be x200 magnification but I do not think this is correct. The cell and nuclei size differ.

Response: I regret that there was an error in writing the manuscript. The magnification scaling was x200 for Figure 1D, and x400 for Figure 1E. Because of your pertinent comments, we corrected the descriptions.  

Figure 2B spelling NLRP3KO SKI should be NLRP3KO AKI

Response: I appreciate your comment. We corrected the spelling error.

Figure 2B – ACSL4 increased in WT AKI. This is also seen in the NLRP3KO AKI but in the text says that there is decreased expression of marker of ferroptotic markers in the NLRP3KO AKI mice compared to the WR AKI.

Response: Thank you for your valuable comment. ACSL4 levels were not different between the KO AKI and WT AKI groups. We erased related comments in the abstract part and corrected the comment in the results.  

In particular, in the kidneys of mice with RIAKI, GPX4 was downregulated, showing that ferroptosis was induced.

Figure 3 C and E, it is difficult to which sets of data on the graphs are significantly different, maybe the size of the graphs should be increased and the size of the western blots should be decreased proportionately.

Response: I appreciate your comment. As you have indicated, the size of the WB pictures was decreased, and that of the graphs was increased proportionally.

Figure 5 B and the legend -  in the figure # is vs control but in the legend # is vs myoglobin+L-AA, and in the Figure *is vs myoglobin+L-AA and in the legend * is vs control.

Response: Thank you for your valuble comment. The * mark indicates the comparison with Mb-untreated cells, and the # mark implies the comparison with siNLRP3-untreated cells. We corrected this information in the legends of Figure 3-5. The marks* and # were exchanged in Figure 5B in the original version, and we switched them. Additionally, we inserted the * mark on the bar of the siNLRP3+Mb group in Figure 4B. 

The discussion is wide and detailed and clearly lays out where this work fits in the field.

In addition to the points you made, we added one reference numbered 24 (KRCP 2018;37(3):183-196) in the discussion part for clarification.